# In Vitro Antimicrobial and Antibiofilm Activity of S-(-)-Limonene and R-(+)-Limonene against Fish Bacteria

**Elisia Gomes da Silva** [1], **Guerino Bandeira Junior** [1], **Juliana Felipetto Cargnelutti** [2], **Roberto Christ Vianna Santos** [3], **André Gündel** [4] and **Bernardo Baldisserotto** [1,*]

1   Department of Physiology and Pharmacology, Universidade Federal de Santa Maria, Santa Maria 97105-900, RS, Brazil; elisia.silva@ufsm.br (E.G.d.S.); guerino.junior@outlook.com (G.B.J.)
2   Department of Preventive Veterinary Medicine, Universidade Federal de Santa Maria, Santa Maria 97105-900, RS, Brazil; jucargnelutti@gmail.com
3   Department of Microbiology and Parasitology, Universidade Federal de Santa Maria, Santa Maria 97105-900, RS, Brazil; robertochrist@gmail.com
4   Department of Physics, Universidade Federal do Pampa, Bagé 96413-170, RS, Brazil; andregundel@unipampa.edu.br
*   Correspondence: bernardo.baldisserotto@ufsm.br; Tel.: +55-55-3220-9382

**Abstract:** Brazilian fish farming goes together with the emergence of numerous bacterial diseases, with *Aeromonas hydrophila* being the main bacterial pathogen. As a consequence, antimicrobials are excessively used. Considering that antimicrobials are relatively stable and nonbiodegradable, medicinal plants and their phytochemicals have been used as alternative treatments of bacteriosis in fish farming. Limonene is a monoterpene available in two enantiomers: R-(+)-limonene and S-(-)-limonene. This study analyzed the antibacterial activity of the phytochemicals S-(-)-limonene and R-(+)-limonene against some bacteria isolated from silver catfish (*Rhamdia quelen*). Furthermore, by means of spectrophotometry and atomic force microscopy, we also investigated the combination therapy of phytochemicals with antimicrobials and their activity in terms of inhibiting biofilm formation. Six clinical isolates and a standard strain were selected for antimicrobial activity testing. Biofilm formation was tested in 96-well plates and nylon cubes. The most sensitive of the strains tested was the *A. hydrophila* strain (MF 372510). S-(-)-limonene and R-(+)-limonene had high minimum inhibitory concentrations; however, they strongly inhibited *A. hydrophila* biofilm formation. R-(+)-limonene and S-(-)-limonene had an additive effect when combined with florfenicol and an antagonistic effect with oxytetracycline. In general, the phytochemicals tested showed strong antibiofilm activity against *A. hydrophila*, and when in combination therapy with florfenicol, they showed an additive effect against the treatment of *A. hydrophila*.

**Keywords:** limonene; *Aeromonas hydrophila*; checkerboard; antibiofilm; atomic force microscopy

## 1. Introduction

Outbreaks of bacteriosis in fish farming are abrupt and cause considerable economic losses [1]. Often, the occurrence of these outbreaks is associated with stressful conditions such as impaired water quality and the handling and transport of fish. Most bacterial pathogens in fish are aerobic and Gram-negative bacteria [2].

*Aeromonas hydrophila* is a Gram-negative bacterium that is considered a potent pathogen in aquatic ecosystems [3–6]. Acute infections with this bacterium usually cause hemorrhagic septicemia, lesions in the mouth and fins, and ulcers progressing to necrosis of the skin and internal organs [7,8]. *Aeromonas veronii* is a species that has genes for resistance to various antimicrobials and has several virulence factors [9]. It is responsible for hemorrhagic septicemia and ulcerative syndrome in fish [10].

*Citrobacter freundii* and *Raoultella ornithinolytica* are Gram-negative bacteria from the Enterobacteriaceae family [11]. Infection with *C. freundii* usually occurs due to poor water quality or immunosuppressed animals [7]. Symptoms include lethargy, disordered

movements, and bleeding [12,13]. *Raoultella ornithinolytica* causes a high production of histamine in fish and is responsible for a toxic condition termed scombroid poisoning, which is related to the ingestion of fish contaminated by the bacterium. The most frequently reported infections by *R. ornithinolytica* are those in the digestive system [14].

*Stenotrophomonas maltophilia* are Gram-negative aerobic bacilli that are considered to be emerging pathogens. It is a bacterium prevalent in aquatic or humid environments and in nosocomial environments and can cause several serious infections in humans [15]. American catfish (*Ictalurus punctatus*) infected with *S. maltophilia* present with infectious intussusception syndrome [16], gilthead seabream (*Sparus aurata* L.) edema, and collapse of the gills [17].

In Brazil, *A. hydrophila* and *A. veronii* are the main species reported [18–20]. The genus *Aeromonas* is capable of forming biofilms and causing hemolysis, which contributes to the virulence of the strain [3,18,20]. Currently, strategies to prevent biofilm formation involve the development of biofilm inhibiting agents with the aim of preventing the early stages of biofilm formation and preventing microcolony formation [21].

Due to the impacts caused by bacterial infections, effective therapeutic options are needed to reduce the morbidity and mortality of animals. However, the indiscriminate use of antimicrobials in fish has shown multiresistance by microorganisms, in addition to the deposition of residues in animal tissues and in the aquatic environment [22,23]. In this context, medicinal plants have been considered a promising alternative in the control and treatment of bacteriosis in fish, either as an isolated therapy or as a combination therapy with antimicrobials [18,24–26]. Through their secondary metabolism, plants give rise to essential oils, which have a set of chemical compounds (phytochemicals) in their composition [27].

Limonene is a monoterpene of secondary metabolism mainly in plants of the genus *Citrus* [28]. It is found in the R-(+)-limonene and S-(-)-limonene isoforms [29]. The R-(+)-limonene isoform is widely used in cosmetic products [30], food [31], and insect control [32].

Several studies have reported that essential oils containing limonene have different biological activities, including antimicrobial activities. Song et al. [33] found that *Citrus reticulata* essential oil, whose composition corresponds to 78% limonene, inhibited the activity of *Staphylococcus aureus*. Haraoui et al. [34] determined the antibacterial activity of *C. max*, *C. aurantium*, and *C. limon* essential oils in several Gram-negative and Gram-positive bacteria. As far as we know, there are no data available on the specific activity of the R-(+)-limonene and S-(-)-limonene isoforms against the bacteria to be tested.

Thus, the aim of this study was to determine the minimum inhibitory concentration (MIC) and the minimum bactericidal concentration (MBC) of the phytochemicals R-(+)-limonene and S-(-)-limonene against *A. hydrophila*, *A. veronii*, *C. freundii*, *R. ornithinolytica*, and *S. maltophilia*, to analyze the combined therapy of limonene enantiomers with florfenicol and oxytetracycline against *A. hydrophila*, and to analyze the inhibition of *A. hydrophila* biofilm formation by means of atomic force microscopy.

## 2. Materials and Methods

### 2.1. Phytochemicals

The phytochemicals R-(+)-limonene and S-(-)-limonene were purchased from Sigma-Aldrich™ (St. Louis, MO, USA).

### 2.2. Minimum Inhibitory Concentration and Minimum Bactericidal Concentration Assays

Six clinical isolates (*A. hydrophila* (GenBank access MF 372509), *A. hydrophila* (MF372510), *A. veronii* (MH 397688), *C. freundii* (MF 565839), *R. ornithinolytica* (MF 372511), and *S. maltophilia* (MT 572493)) and a standard strain (*A. hydrophila* (ATCC® 7966)) were selected for MIC and MBC tests. The MIC and MBC were obtained following the microdilution method from the Clinical and Laboratory Standards Institute (CLSI) guidelines [35], document VET04-A2. S-(-)-limonene and R-(+)-limonene were diluted in 96% ethanol and added to

Mueller–Hinton broth (MHB) at concentrations of 6400, 3200, 1600, 800, 400, 200, 100, 50, 25, 12.5, 6.25, and 3.125 µg/mL (in triplicate).

The inoculum was prepared in saline from cultures grown on Mueller–Hinton agar (MHA) ($1 \times 10^8$ CFU/mL; 0.15—optic density (DO)—600 nm) (28 °C/24 h). Inoculum (10 µL, $1 \times 10^5$ CFU) was added to each well containing the tested substances. The microplates were incubated under aerobic conditions for 24 h at 28 °C. The same procedure was carried out in an ethanol control. Ten microliters of 0.1% resazurin dye (Sigma-Aldrich[TM], Product Code 199303) was added to each well to assist in the MIC reading, which was considered the lowest concentration of the substance that inhibited visible bacterial growth. MBCs were confirmed by reinoculation of 10 µL of each bacterial culture in MHA (28 °C/24 h), and the lowest concentration of antimicrobial that did not show growth was defined as the MBC.

### 2.3. Checkerboard Assay

Different combinations of oxytetracycline and florfenicol with S-(+)-limonene and R-(+)-limonene were tested against the most sensitive strain in the MIC test (*A. hydrophila* MF 372510) using the checkerboard method [36]. The MIC values for florfenicol and oxytetracycline were obtained from Bandeira Junior et al. [18], who used this same bacterial strain and the same methodology. The concentrations tested (in triplicate) in combination were below (MIC/8, MIC/4, MIC/2), equal to (MIC), or above (MIC × 2, MIC × 4, MIC × 8) the MIC for the microorganism tested.

The checkerboard method consisted of lines containing different amounts of Substance A, diluted along the y-axis, and columns containing different amounts of Substance B, diluted along the x-axis. The checkerboard results were analyzed through the lowest fractional inhibitory concentration (FIC) index method, following the Clinical Microbiology Procedures Handbook [37]. The FIC was calculated as follows: FIC of Substance A (FICA) = MIC of Substance A in combination/MIC of Substance A alone; FIC of Substance B (FICB) = MIC of Substance B in combination/MIC of Substance B alone. The FIC index (FICI) was considered to be FICA + FICB. A synergy effect between substances was defined for FICI values less than or equal to 0.5, and additivity was determined for FICI values between 0.5 and 4, while antagonism was determined for FICI values greater than 4 [38].

### 2.4. Effect on Biofilm Formation

Several phytochemical concentrations (MIC/8, MIC/4, MIC/2, MIC, MIC × 2, MIC × 4, and MIC × 8) were tested on *A. hydrophila* biofilms, according to Stepanovic et al. [39].

Cultures were grown overnight in tryptic soy broth (TSB) and diluted to 0.25 OD (600 nm). Subsequently, TSB with the phytochemical to be tested (diluted in ethanol) and bacterial inoculum were added to each well for each concentration tested (in triplicate). After 48 h of incubation at 28 °C, each well was washed (three times), dried, stained with gentian violet, and washed again. The stained biofilms were resuspended in alcohol/acetone (80:20), and the OD (550 nm) was measured with a microplate reader. Controls without inoculum (negative controls) corresponding to each phytochemical were added. Biofilm formation was considered strong when the OD of the sample (ODs) was more than four-fold greater than the OD of the negative control (ODnc), moderate when the ODs was between two- and four-fold greater than the ODnc, and weak when the ODs was up to two-fold greater than the ODnc [40]. In a pilot study, without the addition of phytochemicals, it was observed that the strain tested had a strong ability to form biofilms (data not shown). Consequently, a decrease in the formation of biofilms could be attributed to the addition of phytochemicals.

*2.5. Effect on Biofilm Formation in Nylon Cubes*

The effect of several phytochemical concentrations (MIC/8, MIC/4, MIC/2, MIC, MIC × 2, MIC × 4 and MIC × 8) on the biofilm formation capacity of the *A. hydrophila* (MF 372510) strain was tested in nylon (Braskem, Brazil) rectangular blocks (1.0 × 1.0 × 0.5 cm) previously autoclaved and maintained individually in each well of 24-well flat-bottomed polystyrene microtiter plates.

Cultures in 6 mL of TSB were incubated for 24 h at 28 °C and diluted to 0.25 OD (600 nm). Subsequently, 950 µL of TSB was added, followed by 50 µL of inoculum and 1 mL of TSB with the phytochemicals to be tested in each well (5 replicates for each tested concentration). After 48 h of incubation at 28 °C, each well was washed 3 times with sterile distilled water. After drying, 2 mL of absolute methanol was added to each well and incubated for 1 min. After removing all methanol, the blocks were analyzed using atomic force microscopy.

*2.6. Atomic Force Microscopy*

Images were obtained with an Agilent Technologies 5500 microscope. The images (5 µm × 5 µm) were obtained in noncontact mode using PPP-NCL tips (Nanosensors, force constant = 48 N/m). The images were analyzed with PicoView software.

*2.7. Statistical Analysis*

The Levene test was used to test the homogeneity of variances between the groups. When the data were parametric, groups were compared using unilateral analysis of variance and Tukey's test. In the case of nonparametric data, Kruskal–Wallis ANOVA and multiple comparisons of the mean ratings for all groups were performed (STATISTICA 7.0).

**3. Results**

*3.1. Minimum Inhibitory Concentration and Minimum Bactericidal Concentration Assays*

The S-(-)-limonene and R-(+)-limonene enantiomers showed weak antibacterial activity (MIC 3.2 mg/mL$^{-1}$ and MIC 6.4 mg/mL$^{-1}$, respectively) against the *A. hydrophila* strain (MF 372510) and no activity against *C. freundii*, *R. ornithinolytica*, and *S. maltophilia* (Table 1).

*3.2. Checkerboard Assay*

The results obtained in the checkerboard tests allowed us to infer that limonene can be a good alternative for the treatment of diseases caused by *A. hydrophila*, as an additive effect was demonstrated in 50% of the tested combinations. The phytochemicals S-(-)-limonene and R-(+)-limonene combined with the antimicrobial florfenicol showed an additive effect against *A. hydrophila* (MF 372510), and the same effect presented when the antimicrobials florfenicol and oxytetracycline were combined. The combination therapy of S-(-)-limonene and R-(+)-limonene with each other or in combination with oxytetracycline had an antagonistic effect. None of the combinations tested had a synergistic effect (Table 2).

*3.3. Effect of Biofilm Formation by Optical Microscopy*

The S-(-)-limonene isoform was more effective at inhibiting *A. hydrophila* biofilm formation at MIC × 2, MIC × 4, and MIC × 8, compared to R-(+)-limonene only at MIC × 8 (Table 3).

**Table 1.** Minimum inhibitory concentrations (MICs) and minimum bactericidal concentrations (MBCs) of the optical isomers of limonene against six clinical isolates of pathogenic bacteria from fish and a standard strain.

| | *Aeromonas hydrophila* ATCC® 7966 | | *Aeromonas hydrophila* MF 372509 | | *Aeromonas hydrophila* MF 372510 | | *Aeromonas veronii* MH 397688 | | *Citrobacter freundii* MF 565839 | | *Raoultella ornithinolytica* MF 372511 | | *Stenotrophomonas maltophilia* MT 572493 | |
|---|---|---|---|---|---|---|---|---|---|---|---|---|---|---|
| | MIC mg mL$^{-1}$ | MBC mg mL$^{-1}$ | MIC mg mL$^{-1}$ | MBC mg mL$^{-1}$ | MIC mg mL$^{-1}$ | MBC mg mL$^{-1}$ | MIC mg mL$^{-1}$ | MBC mg mL$^{-1}$ | MIC mg mL$^{-1}$ | MBC mg mL$^{-1}$ | MIC mg mL$^{-1}$ | MBC mg mL$^{-1}$ | MIC mg mL$^{-1}$ | MBC mg mL$^{-1}$ |
| **S-(−)-limonene** | >6.4 | >6.4 | >6.4 | >6.4 | 3.2 | 3.2 | >6.4 | >6.4 | >6.4 | >6.4 | >6.4 | >6.4 | >6.4 | >6.4 |
| **R-(+)-limonene** | >6.4 | >6.4 | >6.4 | >6.4 | 6.4 | 6.4 | >6.4 | >6.4 | >6.4 | >6.4 | >6.4 | >6.4 | >6.4 | >6.4 |

**Table 2.** Fractional inhibitory concentration (FIC) and fractional inhibitory concentration index (FICI) of phytochemicals S-(+)-limonene (SL) and R-(+)-limonene (RL) in association with different combinations of the antimicrobials florfenicol (FLF) and oxytetracycline (OXT) against *Aeromonas hydrophila* MF 372510.

| DRUGS | FIC | FICI |
|---|---|---|
| SL—FLF | | |
| SL | 0.125 | 2.125 [b] |
| FLF | 2 | |
| SL—OXT | | |
| SL | 0.125 | 4.125 [c] |
| OXT | 4 | |
| RL—FLF | | |
| RL | 0.125 | 2.125 [b] |
| FLF | 2 | |
| RL—OXT | | |
| RL | 0.125 | 4.125 [c] |
| OXT | 4 | |
| SL—RL | | |
| SL | 4 | 8 [c] |
| RL | 4 | |
| FLF—OXT | | |
| FLF | 1 | 2 [b] |
| OXT | 1 | |

[b] additivity; [c] antagonism.

**Table 3.** Effects of phytochemicals S-(+)-limonene and R-(+)-limonene on the biofilm formation of *Aeromonas hydrophila* isolated from fish (MF 372510).

| | *Aeromonas hydrophila* **MF 372510** | |
|---|---|---|
| | **S-(-)-limonene** | **R-(+)-limonene** |
| MIC × 8 | 0 | 1 |
| MIC × 4 | 0 | 3 |
| MIC × 2 | 1 | 3 |
| MIC | 3 | 3 |
| MIC/2 | 3 | 3 |
| MIC/4 | 3 | 3 |
| MIC/8 | 3 | 3 |

MIC, minimum inhibitory concentration; 0, no biofilm production; 1, weak biofilm production; 2, moderate biofilm production; 3, strong biofilm production.

### 3.4. Effect of Biofilm Formation by Atomic Force Microscopy

Figure 1 shows the atomic force microscopy (AFM) results on nylon rectangular block surfaces in the control groups (Figure 1A), S-(-)-limonene (Figure 1B) and R-(+)-limonene (Figure 1C) results at low concentrations, and S-(-)-limonene (Figure 1D) and R-(+)-limonene (Figure 1E) results at high concentrations.

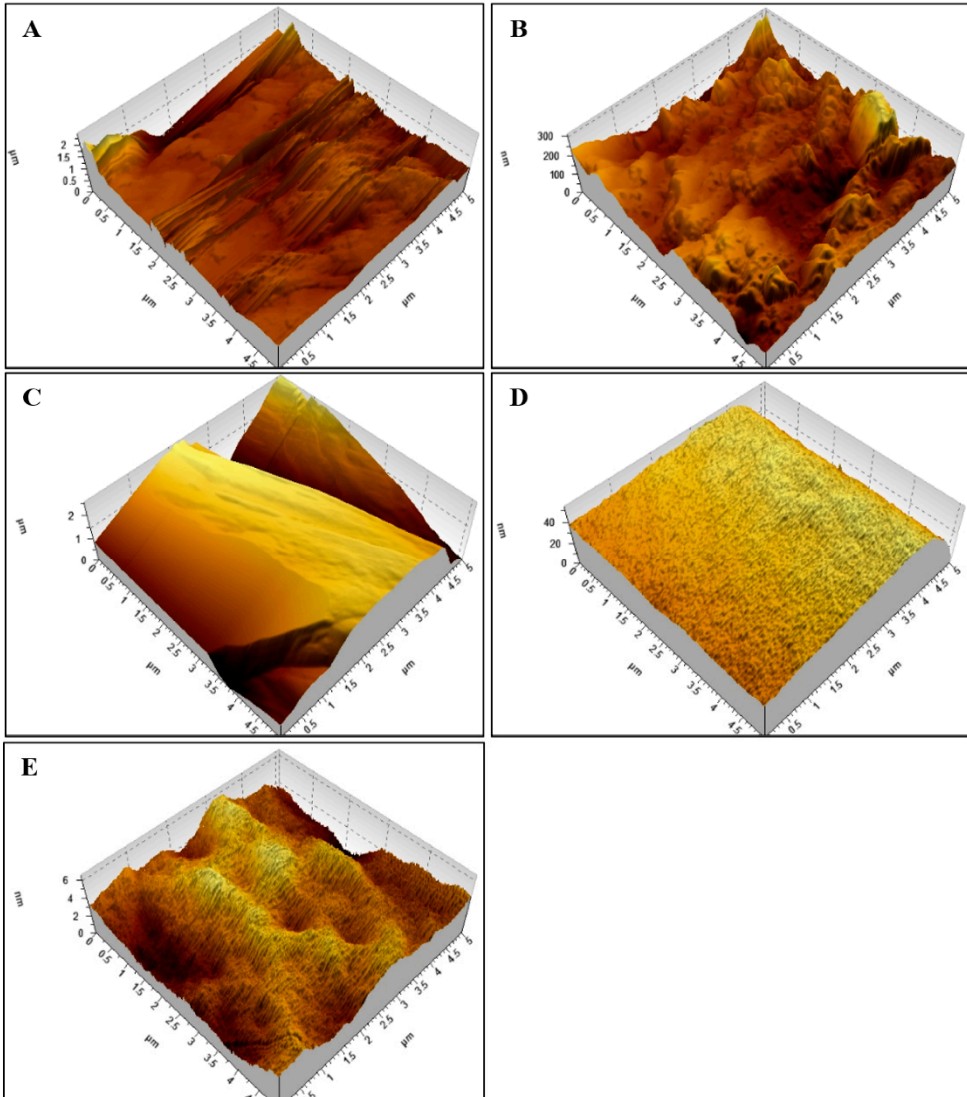

**Figure 1.** Atomic force microscopy (AFM) on a nylon surface. Comparative images between control
—2 μm (**A**), low concentrations (minimum inhibitory concentration—MIC/8, MIC/4, and MIC/2):
S-(+)-limonene—0.3 μm (**B**) and R-(+)-limonene—0.8 μm (**C**), high concentrations (MIC × 2, MIC × 4,
and MIC × 8): S-(+)-limonene—0.006 μm (**D**), and R-(+)-limonene—0.04 μm (**E**).

Figure 2 shows high peaks of roughness (2 μm) in the control group, indicating the
formation of *A. hydrophila* biofilm. In the S-(-)-limonene and R-(+)-limonene groups, the
lower concentrations of these isomers significantly decreased the formation of *A. hydrophila*
biofilms (0.3 μm and 0.8 μm, respectively) compared to the control group. The higher
concentrations of S-(-)-limonene (0.006 μm) and R-(+)-limonene (0.04 μm) also statistically
reduced the biofilm formation of *A. hydrophila*, when compared to the control.

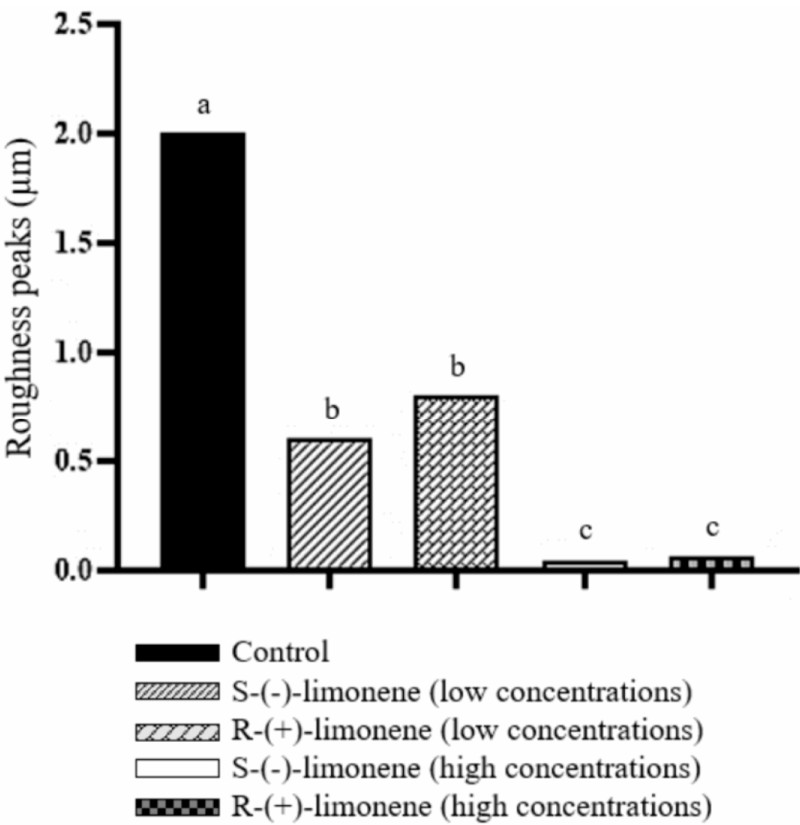

**Figure 2.** Effect of S-(-)-limonene and R-(+)-limonene on the inhibition of *A. hydrophila* biofilms in rectangular nylon blocks. Data are shown as the mean ± standard error. One-way analysis of variance and Tukey test, $p < 0.05$. Distinct letters indicate a significant difference between treatments.

## 4. Discussion

Brazilian fish farming is growing, and allied with this expansion, numerous bacterial diseases can arise [41]. Consequently, the indiscriminate use of antimicrobials may increase [22]. The occurrence of multiresistant bacteria in aquatic environments has been constant [42], a fact that has driven research into medicinal plants and their major compounds as alternatives to antimicrobials [25,26].

When studying limonene enantiomers, Lis-Balchin et al. [43] found that R-(+)-limonene exhibited greater activity against several bacterial species and different strains of *Listeria monocytogenes* than S-(-)-limonene. Lee et al. [44] verified the significant role of S-(-)-limonene in suppressing the growth of the bacterium *Xanthomonas oryzae* pv. *Oryzae*, and using extracts of orange and lemon peel against the fungi *Candida albicans*, *Aspergillus niger*, *Aspergillus* sp., and *Penicillium* sp., Omran et al. [45] verified that the S-(-)-limonene present in lemon peel had a greater inhibitory effect on the examined fungi than R-(+)-limonene.

In the MIC and MBC tests, the S-(-)-limonene and R-(+)-limonene isoforms showed a weak minimum inhibitory concentration for the *A. hydrophila* strain (MF 372510), showing no activity for the other tested bacteria. However, the MIC was equal to the MBC, suggesting that these phytochemicals can be bacteriostatic and bactericidal at the same concentration, as also described by Pathirana et al. [46] for an essential oil of *Citrus aurantifolia* and limonene against pathogenic bacteria of sole *Paralichthys olivaceus*. Cunha et al. [24] reported that most essential oils, including their isolated compounds and even those that have an MIC that is greater than that recommended by Ríos and Recio [47], are effective at treating bacterial infections in fish at concentrations below the MIC observed in vitro.

Few countries monitor the amounts of antimicrobials used in fish farming, which demonstrates the scarcity of data. In Brazil, oxytetracycline and florfenicol are the legal antimicrobials for use in aquaculture [48,49]. Oxytetracycline is one of the main antimicro-

bials recently legalized to treat bacterial diseases in some aquatic species [48]. It has low tissue absorption and distribution in fish, which consequently maximizes the use of high doses and can favor multiresistance by microorganisms [50]. Florfenicol, in addition to being the first and main choice for the treatment of bacteriosis in Brazil [48] and in several countries [51,52], has a broad spectrum of action and is bacteriostatic in the treatment of bacterial diseases in fish [53].

There have been some studies that have shown the susceptibility of several strains of *A. hydrophila*, *A. veronii*, *C. freundii*, and *R. ornithinolytica* to florfenicol and oxytetracycline. Janda and Abbott [4] reported the low susceptibility of *Aeromonas* sp. to various classes and combinations of antimicrobials, in addition to their ability to develop resistance to antimicrobials in the aquatic environment. Bandeira Junior et al. [18] showed the resistance of some strains of *A. hydrophila*, *C. freundii*, and *A. veronii* to florfenicol and oxytetracycline. However, the same authors verified that when the therapy of these antimicrobials was combined with the phytochemicals tested, therapeutic success was found in all the bacteria tested, suggesting that this combination resulted in activity or synergy.

The combined therapy of one or more antimicrobials with phytochemicals has been an effective alternative treatment of a disease, as well as a means to reduce the amounts of antimicrobials used, to reduce the speed of development of multiresistance, and to protect the environment from waste contamination [54,55]. Our checkerboard results suggest that a combined therapy of the R-(+)-limonene and S-(-)-limonene enantiomers with florfenicol for treatment against *A. hydrophila* is possible, but combined therapy with oxytetracycline is not possible due to the presented antagonistic effect.

In this investigation, the combination of florfenicol and oxytetracycline produced an additive effect, the same effect shown by the limonene enantiomers. Considering the increase in environmental and health problems due to the indiscriminate use of antimicrobials and the safety regarding the biodegradability and lesser toxicity of essential oils and their compounds in mammals, birds, and fish [56,57], a combined therapy of limonene enantiomers with florfenicol is significantly safer and less expensive.

It is important to emphasize that the biological activity of a chiral substance can vary according to its stereoisomerism, and although they have the same chemical structure, when in a racemic mixture, the enantiomers are also capable of demonstrating different biological responses [58]. Some studies have reported that pure enantiomers are generally less biologically active since greater activity is often associated with the racemic mixture due to the interaction of the components [59]. Vuuren and Viljoen [60] reported that the racemic mixture of limonene had a better FIC than the isolated enantiomers against the tested bacteria. Contrary to these results, our study found that the combination of the enantiomers R-(+)-limonene and S-(-)-limonene should be avoided since it presents an antagonistic effect in the treatment of *A. hydrophila*.

Approximately 65% of all bacterial infections are related to bacterial biofilms [61]. Biofilms are ecosystems of microorganisms fixed on a surface, formed by one or more species, surrounded by a matrix of exopolysaccharides [62]. As biofilms mature and multiply, microcolonies appear and are responsible for inducing the formation of channels, which favor the transfer of nutrients, oxygen, and mainly genetic material such as plasmids, which are important for the mechanisms of resistance to antimicrobials and virulence [63].

There are few reports on the antimicrobial activity of isoforms isolated from S-(-)-limonene and R-(+)-limonene. In the analysis of the inhibition of biofilm formation through spectrophotometry, the S-(-)-limonene isoform was more effective in the antibiofilm activity of *A. hydrophila* than R-(+)-limonene. There are no data on the antibiofilm activity of the isolated S-(-)-limonene and R-(+)-limonene enantiomers. Subramenium et al. [64] showed the antibiofilm activity of limonene against different species of *Streptococcus* sp.

Different microscopic techniques are employed for the analysis of biofilms. AFM has been proven to be a reliable tool for studying films embedded in a matrix, such as biofilms [64–66]. In assessing the inhibition of biofilm formation through AFM, we used nylon as a substrate, which is the main polymer used in making net tanks in fish farming.

Some factors, such as the nature of the substrates, interfere and favor the formation of biofilms at different stages [67]. Cai and Arias [68] proved the ability of *A. hydrophila* to form biofilms on the surfaces of all materials tested in aquaculture facilities. The two isoforms of limonene inhibited the formation of *A. hydrophila* biofilms in nylon cubes. The inhibition of biofilm formation is a fundamental step in decreasing the pathogenic effect of bacteria [69]. Similar results were also found with *Citrus limonum*, the composition of which corresponds to 70% limonene, against biofilms of *Klebsiella* sp. [70]. In addition, the results of a surface coating test suggested that the inhibition of bacterial adhesion to surfaces could be the mode of action of limonene, thus avoiding the cascade of biofilm formation [65].

The results obtained in this study show that, although S-(-)-limonene and R-(+)-limonene had a weak minimal inhibitory concentration, they strongly inhibited the biofilm formation of *A. hydrophila* in the tests performed. An analysis of the MIC and the minimum effective concentrations of essential oils, as well as their major compounds used in in vivo tests demonstrated that there was no correlation between them. Consequently, the value of the MIC in vitro does not allow for a good prediction of the in vivo effect of an essential oil [24].

In conclusion, we propose the investigation of the antibacterial activity of these phytochemicals against bacteriosis in vivo, as well as for the prevention of contamination linked to the formation of biofilms, as we found satisfactory results in the reduction of nylon biofilms, which would justify their use in the treatment/prevention of biofilms in networks. The phytochemicals tested may be promising combination therapy agents with florfenicol for potential applications in clinical fish infections.

**Author Contributions:** E.G.d.S.: conceptualization, methodology, validation, formal analysis, investigation, resources, data curation, writing—original draft, writing—review and editing, visualization. G.B.J.: methodology, writing—review and editing. J.F.C.: writing—review and editing. R.C.V.S.: methodology, writing—review and editing. A.G.: writing—review and editing. B.B.: conceptualization, methodology, validation, formal analysis, investigation, resources, data curation, writing—review and editing, supervision, project administration, funding acquisition. All authors have read and agreed to the published version of the manuscript.

**Funding:** Aperfeiçoamento de Pessoal de Nível Superior (CAPES, Brazil)—Finance Code 001. Conselho Nacional de Desenvolvimento Científico e Tecnológico (CNPq, Brazil)—Finance Code 301225/2017-6.

**Institutional Review Board Statement:** Not applicable.

**Informed Consent Statement:** Not applicable.

**Data Availability Statement:** All data are contained within the article.

**Acknowledgments:** B. Baldisserotto and R.C.V. Santos received research fellowships from the Conselho Nacional de Desenvolvimento Científico e Tecnológico (CNPq, Brazil). G. Bandeira Junior received a PhD scholarship from the Coordenação de Aperfeiçoamento de Pessoal de Nível Superior (CAPES, Brazil), Finance Code 001.

**Conflicts of Interest:** The authors declare that they have no conflict of interest.

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
