# Peer review of "In Vitro Antimicrobial and Antibiofilm Activity of S-(-)-Limonene and R-(+)-Limonene against Fish Bacteria"

_fishes, doi:10.3390/fishes6030032_

Round 1

Reviewer 1 Report

This is a straightforward work that talks about mitigating  bacterial pathogens isolated from  silver catfish using R-(+)-limonene and S-(-)-limonene.

Major comment

I would suggest to test the antibiofilm activity of limonene against polybiofilm/mixed biofilm consortium of the bacteria isolated form the fish. Because I think in ecology and in fishes, the bacteria tend to coexist as mixed biofilms and the compound should mitigate them if they are to be considered efficient.

MInor corrections:

1.CV biofilm quantification data should be provided in the main figure panels.

2. The result texts  are very less. Authors should elaborate.

3. The conclusion in the abstract should be modified.

Author Response

Dear Reviewer,

Please see the suggested requests in the attachment.

Reviewer 2 Report

It was with pleasure that I read the manuscript: In vitro antimicrobial and antibiofilm activity of S-(-)-limonene and R-(+)-limonene against fish bacteria, although no significant antimicrobial activity was achieved, the potential as antibiofilm is also very important. I believe that other strategies to detect antimicrobial or antibiofilm could have been applied since in " in vitro" the conditions tested are not equal as in "in vivo", maybe other outcomes could have come.

Also, a in vivo trial seems viable, at least at a small scale to validate the antibiofilm activity of limonene. The manuscript is well written and well presented.

Author Response

(The authors gave the same response as above.)

Round 2

Reviewer 1 Report

The authors have addressed the comments.